# Analysis of the Response Characteristics of Toluene Gas Sensors with a ZnO Nanorod Structure by a Heat Treatment Process

**DOI:** 10.3390/s22114125

**Published:** 2022-05-29

**Authors:** Dae-Hwan Kwon, Eui-Hyun Jin, Dae-Hwang Yoo, Jong-Wook Roh, Dongjun Suh, Walter Commerell, Jeung-Soo Huh

**Affiliations:** 1Korea Gas Safety Corporation, Wonjung-ro, Maengdong-myeon, Eumseong-gun 27738, Chungcheongbuk-do, Korea; chocopy@kgs.or.kr; 2School of Convergence and Fusion System Engineering, Kyungpook National University, 2559, Gyeongsang-daero, Sangju-si 37224, Gyeongsangbuk-do, Korea; eddiejins@naver.com (E.-H.J.); dongjunsuh@knu.ac.kr (D.S.); 3Institute for Global Climate Change and Energy, Kyungpook National University, Sankyuk-dong, Puk-Gu, Daegu 41566, Korea; dhyoo@knu.ac.kr; 4School of Nano and Materials Science and Engineering, Kyungpook National University, 2559, Gyeongsang-daero, Sangju-si 37224, Gyeongsangbuk-do, Korea; jw.roh@knu.ac.kr; 5Technische Hochschule Ulm, Eberhard-Finckh-Strasse 11, 89075 Ulm, Germany; walter.commerell@thu.de

**Keywords:** heat treatment, ZnO nanorod, metal oxide semiconductor gas sensor, toluene gas

## Abstract

The sensing characteristics of toluene gas are monitored by fabricating ZnO nanorod structures. ZnO nanostructured sensor materials are produced on a Zn film via an ultrasonic process in a 0.01 M aqueous solution of C_6_H_12_N_4_ and Zn(NO_3_)_2_∙6H_2_O. The response of the sensors subjected to heat treatment in oxygen and nitrogen atmospheres is improved by 20% and 10%, respectively. The improvement is considered to be correlated with the increase in grain size. The relationship between the heat treatment and sensing characteristics is evaluated.

## 1. Introduction

Pollution and toxic gases, such as toluene, in plants and at home, in addition to outdoors, pose occupational and health hazards. Thus, gas-sensing devices need to be installed in such places [1,2,3,4,5,6].

In the past decade, metal-oxide gas detectors have attracted substantial interest owing to their low cost, flexible production, simple application, a high number of detectable gases, and potential for integration with semiconductors [7,8]. Various metal-oxide-based materials are used for gas detection because of their numerous advantages, such as good response characteristics [9,10,11].

The function of the semiconductor-type gas sensor is based on the surface change that occurs during electron exchange with the gas when gas molecules are adsorbed on the surface of the oxide semiconductor at high temperatures (200–400 °C) and change in electrical conductivity [12]. Semiconductor sensors with nanostructures have a surface area adsorbing as much of the target gas as possible, yielding stronger and more measurable response characteristics (particularly at low concentrations) [13]. ZnO is a promising material (II–VI compound semiconductor) having a hexagonal crystal structure referred to as wurtzite, direct-transition bandgap energy of 3.37 eV, and exciton binding energy of 60 meV [14,15]. Nanostructures, such as nanorods and nanowires, are among the main factors that improve the gas-sensing properties of ZnO. In a nanorod structure having a relatively high surface-area-to-volume ratio, charge exchange actively proceeds in the oxidation–reduction process, and the electron mobility and charge aggregation are improved [16,17,18].

Several studies have been conducted to analyze the effect of annealing on the quality of the ZnO nanostructure. Muchuweni et al. reported that the structural properties of a ZnO thin film fabricated by a hydrothermal synthesis method were improved after heat treatment, but the optical properties were inferior [19]. Suresh Kumar et al. investigated the effect of heat treatment of the ZnO seed layer at various temperatures on the change in ZnO nanocrystals [20]. In this study, we develop a ZnO nanostructured sensor that can be used for a toluene gas-sensing device installed in a wastewater treatment plant and analyze its response characteristics. To manufacture the gas sensor, an alumina substrate was coated with Pt and Zn film as an electrode and seed layer on each side, respectively. The Pt electrode and Zn film were fabricated by ion plasma and direct-current sputtering methods, respectively. To turn the deposited zinc film into a zinc oxide film used for the seed layer, the gas sensors were heat-treated at 600 °C for 1 h (primary heat treatment). Gas sensors of ZnO nanorods were then manufactured by an ultrasonic process in an aqueous solution [21,22,23]. An additional heat treatment (secondary heat treatment) was performed in oxygen and nitrogen atmospheres for 1 h to analyze the sensor’s response characteristics after the secondary heat treatment. The response characteristics of the sample subjected to the secondary heat treatment were compared to the characteristics of the sample subjected to the primary heat treatment. Through an X-ray diffraction (XRD) analysis, we verified the change in the crystallite size of the ZnO nanostructures under different annealing conditions.

## 2. Materials and Methods

As shown in Figure 1, the dimensions of the sensor’s substrate are 4.5 mm × 3.78 mm, while its thickness is 0.3 mm. It consists of gold electrodes and an Al_2_O_3_ substrate. The resistance of the platinum heater (backside) is approximately 15 Ω. To prevent short circuits, the electrodes have positioned holes in the semiconductor gas sensors, as presented below.

Figure 2 shows the sample preparation process. In the pretreatment, as it is not bonded to the zinc seed layer on the Al_2_O_3_ substrate, a platinum film was coated by an ion coater as a bonding layer. The thickness of the platinum film was approximately 80 Å. The Zn membrane with a thickness of 1000 Å was vacuum-metalized by a sputtering machine using metallic zinc as the seed layer. To obtain zinc oxide, the substrates were heat-treated at 600 °C for 1 h in a furnace (primary heat treatment). The substrates were treated with a dissolving solution of zinc nitrate hydrate [Zn(NO_3_)_2_·6H_2_O] and hexamethylenetetramine [C_6_H_12_N_4_] in deionized water. To help the nanostructure formation, the solution was stirred for 1 h by a stirrer. In the sonication procedure, ZnO nanostructures were grown by a sonicator in the prepared solution. An additional heat treatment (secondary heat treatment) was performed in oxygen and nitrogen atmospheres for 1 h to analyze the sensor’s response characteristics after the secondary heat treatment. For the heat treatment, an atmosphere furnace was used. At this time, the rate of gas flow into the inside was 2 L/min. The internal pressure was maintained at 1 atm.

Based on the results of this study, five groups of gas sensors were manufactured with different ultrasonic treatment energies of 50,000, 150,000, 200,000, 250,000, and 300,000 J, denoted as U_5_, U_15_, U_20_, U_25_, and U_30_, respectively. The measuring system for toluene gas detection consisted of a power supply, mass-flow controller, data acquisition board, air, target gas, and circuit board. An experiment was conducted at 350 °C to analyze the response characteristics for gas sensing. Toluene gas (concentration of 100 ppm) was diluted to 20 ppm with dry air and flowed at a rate of 250 sccm.

The resistance of the sensor was measured. The response (*S*) was obtained by evaluating the ratio of the resistance change of the sensor after gas injection to the resistance before gas injection:(1)$$S\;\left(\%\right)=\frac{R_{gas}-R_{air}}{R_{air}}\times 100.$$

## 3. Results and Discussion

### 3.1. Field-Emission Scanning Electron Microscopy (FESEM) of the Pretreated Sensors

Figure 3 shows FESEM (Hitachi SU8220) images of the morphologies of the ZnO sensors. U_5_ and U_15_ exhibited nanostructures composed of nanoparticles, while U_20_, U_25_, and U_30_ exhibited nanostructures with nanorods. This indicates that it is possible to control the shape of the nanostructure by varying the ultrasonic energy, and that the shape of the nanostructure is changed from nanoparticle to nanorod with the increase in energy. Jung et al. reported that the length of the ZnO nanorod arrays can be controlled by the ultrasonication time [21]. In the case of U_20_, U_25_, and U_30_, the length and size of the nanorods increased with the ultrasonic energy, consistent with the results of Jung et al.

### 3.2. Analysis of the Pretreated ZnO Nanostructures

Figure 4 shows the response (%) and recovery (%) of the ZnO nanostructure toxic gas sensors for 20 ppm toluene gas. When toluene gas molecules contact ZnO, they adsorb onto the sensor surface and react with the oxygen ions, which are chemisorbed on the ZnO surface and emit electrons back to the conduction band of ZnO, leading to a potential barrier to charge transport. The gas-sensing properties are strongly dependent on the effective extent of active sites on the ZnO surface [24,25].

The responses of U_5_ and U_15_ having nanoparticle structures were approximately 40 ± 2%. In the case of U_20_, U_25_, and U_30_ having nanorod structures, the response increased from 50 ± 2% to 64 ± 3%. This implies that the sensing characteristics of the sensor having the nanorod structure are better than those of the sensor having the nanoparticle structure. The response was improved to some extent as the surface area of the nanorods increased. In contrast, the recovery of the gas sensor with nanoparticle structures was better than that of the sensor with the nanorod structures. This is likely because, in the case of the nanoparticle structure, more molecules simply stay on the surface than gas molecules penetrating deep inside, and the adsorption length is smaller so that they can be removed relatively easily compared to the nanorod structure.

### 3.3. Analysis of the Response Characteristics of the Sensors

The heat-treatment effect on the gas sensor characteristics was investigated by heat-treating the U_30_ sensor with the highest response in oxygen and nitrogen atmospheres. The sensors heat-treated in oxygen and nitrogen atmospheres were denoted as U_30-O_ and U_30-N_, respectively. Figure 5 shows the response and recovery characteristics of the U_30_, U_30-N_, and U_30-O_ sensors for 20 ppm toluene gas. The response of the U_30_, U_30-N_, and U_30-O_ sensors was 64 ± 3%, 70 ± 3%, and 80 ± 4%, respectively. Thus, an enhancement of 16% was observed between the U_30_ and U_30-O_ sensors. At 400 °C, the toluene gas enabled injecting more electrons into the depleted layer of ZnO nanorods, which was effective even if the heat treatment was performed in a nitrogen atmosphere. In particular, it is considered that the amount of oxygen chemically adsorbed on the ZnO surface increases by the heat treatment in an oxygen atmosphere, thereby improving the reactivity with the toluene gas. Contrary to the sensing properties, the recovery properties were weakened for the heat-treated sensor. It is considered that the high reactivity of the heat-treated sensor surface slightly interferes with the recovery properties.

To verify the ZnO nanostructure of the samples, U_30_, U_30-N_, and U_30-O_, the XRD (Rigaku D/Max-2500) characteristics of each sample were investigated, as shown in Figure 6. The peaks corresponding to the (110), (101), (110), (103), and (112) directions were observed.

The peak value of the secondary-heat-treated sensor was increased. The crystallite size of ZnO was also increased. According to the XRD data, the grain sizes of the ZnO nanorods were calculated (Table 1). To calculate the grain size (*D*) of the ZnO nanostructure, the Scherrer’s equation was used:(2)$$D=k\;\frac{\lambda }{\beta \;\cos\theta },$$
where *k* is a shape factor (0.94), *λ* is the X-ray wavelength (1.5406 Å), *β* is the line broadening at half of the maximum intensity (full width at half maximum (FWHM)) obtained after subtracting the instrumental line broadening, and *θ* is the Bragg angle. The grain size of the U_30_, U_30-N_, and U_30-O_ sensors was 190 nm, 200 nm, and 208 nm, respectively. Thus, the grain size increased by approximately 5% and 10% upon the heat treatment. Thus, according to the XRD analysis, it is considered that more electrons can penetrate the depleted layer of the ZnO surface, which can be related to the increase in grain size through the heat treatment [23].

The reproducibility of the response characteristics of the sensors was examined, as shown in Figure 7. In this test, 20 ppm toluene gas was injected for 5 min, and the sensors were restored in the air for 15 min. This was repeated five times. During this process, the three sensors exhibited good reproducibility; the higher response of the U_30-O_ sensor was maintained.

The sensing properties for various concentrations of toluene are shown in Figure 8. In this experiment, the toxic gas response and restoring process were repeated five times with changed concentrations to 1, 5, 10, 30 and 50 ppm. Toluene gas was injected for 2 min, and the sensors were restored for 5 min. Each sensor exhibited good detection characteristics in proportion to the toluene concentration of 1 to 50 ppm. They exhibited excellent sensing characteristics, even in the low concentration range of 1 ppm. In particular, the sensor heat-treated in the oxygen atmosphere exhibited the best characteristics and can be used as a toluene gas sensor in a wide concentration range.

## 4. Conclusions

In this study, the sensing characteristics of toluene gas were investigated by fabricating sensors having a ZnO nanoparticle structure and nanorod structure. The sensor with the nanorod structure exhibited better sensing characteristics than those of the sensor with the nanoparticle structure. After the nanorod-structured sensor was heat-treated in oxygen and nitrogen atmospheres, the sensing characteristics were improved. A macrocapable state with a higher penetrability was obtained, which could be correlated with the increase in grain size through the XRD analysis. In the future, the structural characteristics of heat treatment will be investigated, and the relationship between the heat treatment and the improvement in sensing characteristics will be clarified.

## Figures and Tables

**Figure 1 sensors-22-04125-f001:**
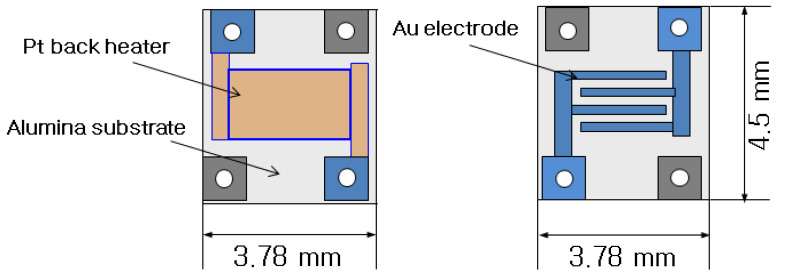
Structures of the fabricated sensor’s substrate.

**Figure 2 sensors-22-04125-f002:**
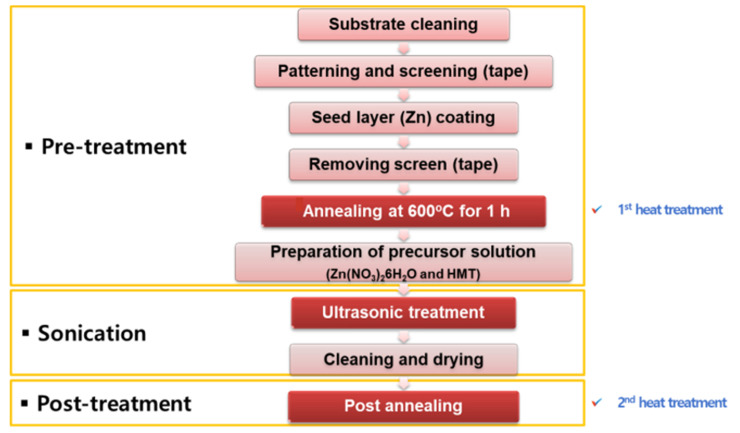
Schematic of the experiment.

**Figure 3 sensors-22-04125-f003:**
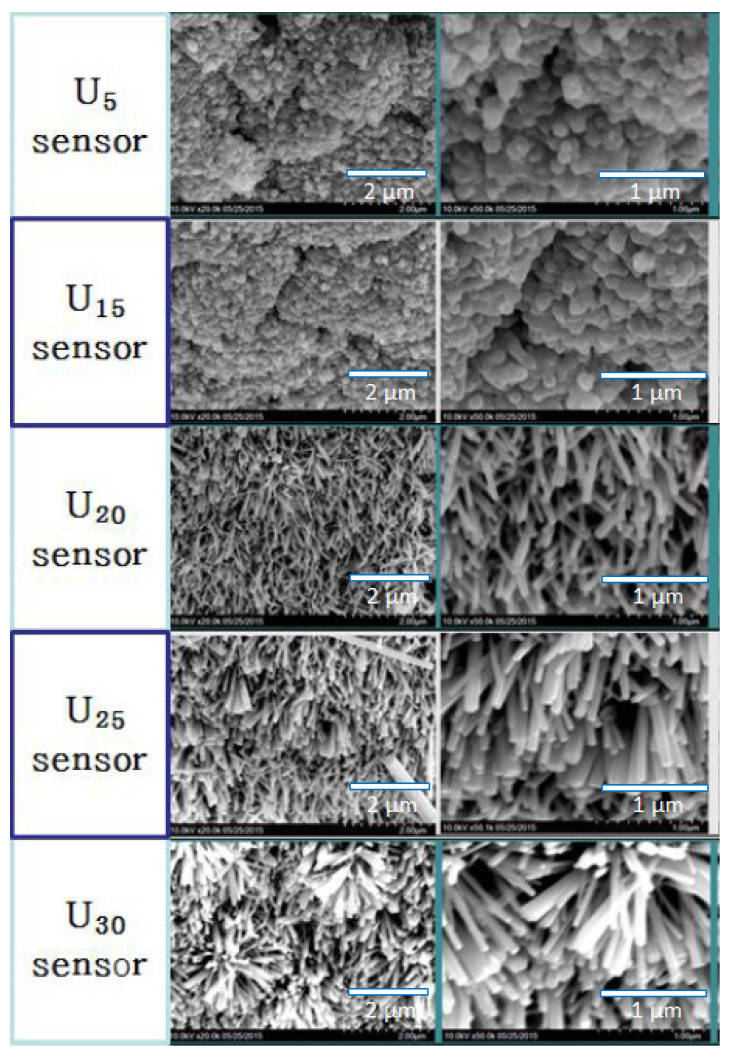
FESEM images of the ZnO nanostructures.

**Figure 4 sensors-22-04125-f004:**
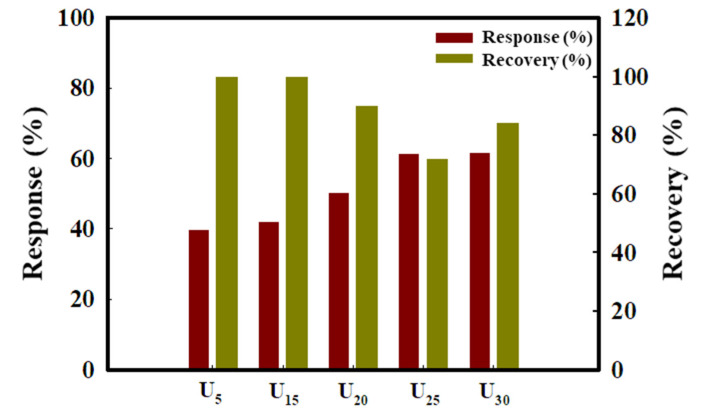
Response and recovery of the ZnO nanostructure sensors for 20 ppm toluene gas.

**Figure 5 sensors-22-04125-f005:**
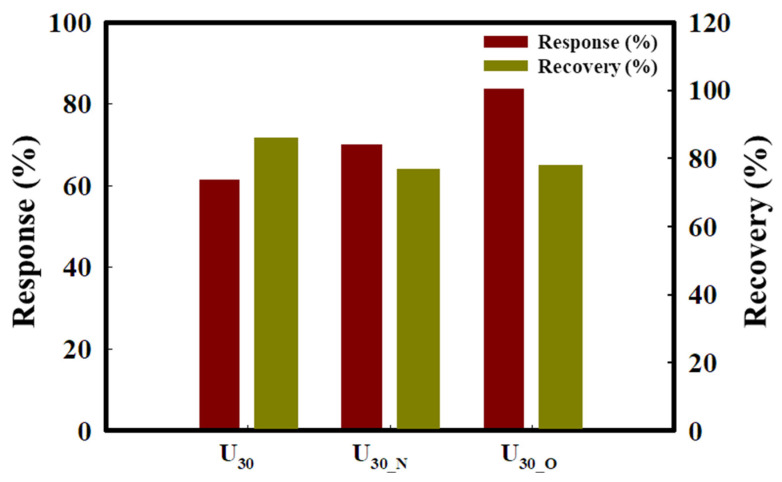
Response and recovery properties of U_30_, U_30-N_, and U_30-O_ before and after heat treatment for 20 ppm toluene gas.

**Figure 6 sensors-22-04125-f006:**
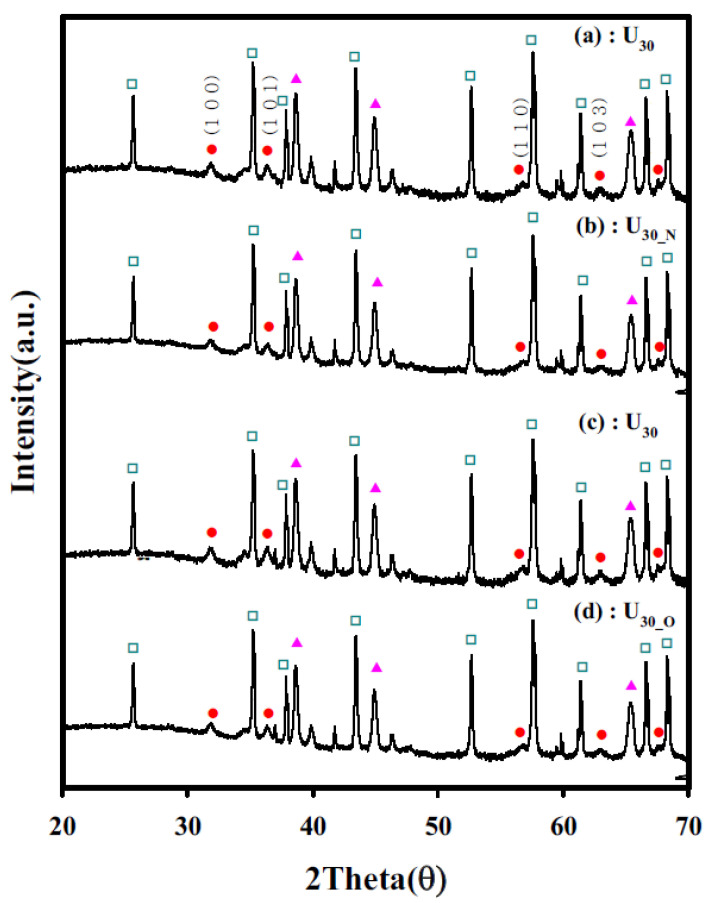
Comparison of XRD patterns of the original and additionally heat-treated sensors (
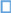
: Al2O3, 
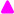
: Pt, 
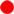
: ZnO peak, (**a**,**c**) U_30_ sensor; (**b**) U_30-N_ sensor; (**d**) U_30-O_ sensor).

**Figure 7 sensors-22-04125-f007:**
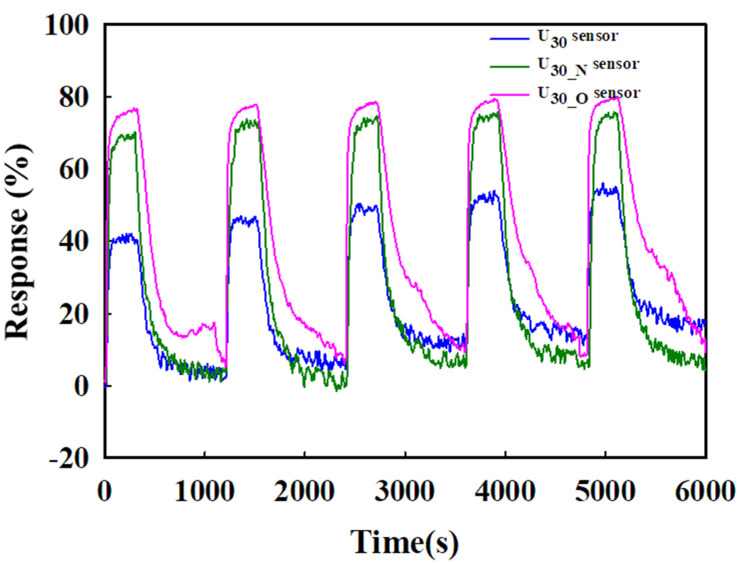
Reproducibility evaluation of U_30_, U_30-N_, and U_30-O_ against 20 ppm toluene.

**Figure 8 sensors-22-04125-f008:**
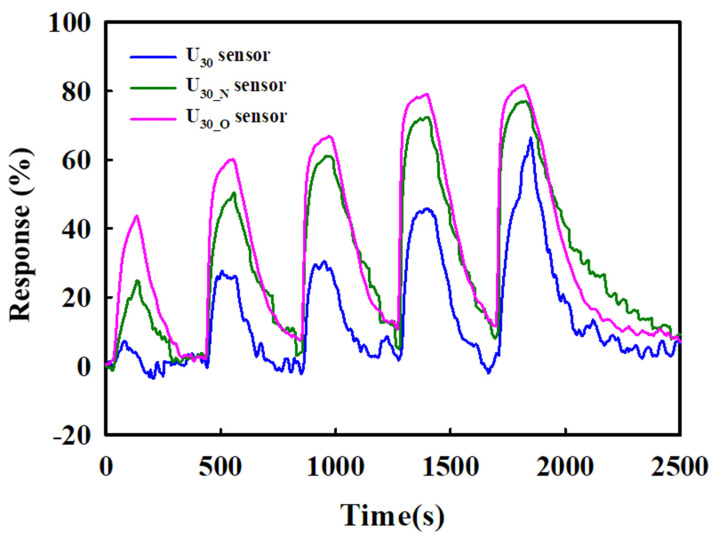
Response characteristics of U_30_, U_30-N_, and U_30-O_ for 1, 5, 10, 30, and 50 ppm toluene.

**Table 1 sensors-22-04125-t001:** Grain sizes of the original and additionally heat-treated sensors.

Sample	Wavelength (nm)	2*θ* (^o^)	FWHM	Crystallite Size (nm)
(a), (c): U_30_	0.154	31.8246	0.4343	190
(b): U_30-N_	0.154	31.8168	0.4172	200
(d): U_30-O_	0.154	31.8299	0.3967	208

## Data Availability

The data that support the findings of this study are available from the corresponding authors upon request.

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
