# Peer review of "Analysis of the Response Characteristics of Toluene Gas Sensors with a ZnO Nanorod Structure by a Heat Treatment Process"

_sensors, 2022, doi:10.3390/s22114125_

Round 1
Reviewer 1 Report
More discussion should be added about the ultrasonic treatment for nanostructure growth - what is the mechanism? Why does the increase in ultrasonic energy result in longer nanorods? At least add references to prior art for reference if not able to add complete discussion.
In the outline of chemical reactions underlying the sensing mechanism on page 5, there is so much detail about the oxygen adsorption and then the toluene reaction is given one line. Toluene will not decompose to 7 carbon dioxide molecules all at once. If the authors are arguing that the catalytic decomposition of toluene to CO2 is responsible for the sensing mechanism, then more discussion of the catalysis would be useful. However, it's not clear that this is the true mechanism - preferential binding of toluene to the semiconductor displacing surface oxygen is the more commonly understood mechanism.
There are no controls shown for the sensing experiments. Response of heated bare substrate (prior to Pt and Zn deposition) should be shown to ensure that the microheater platform itself is not contributing to the sensor signal. How was the temperature of 350 C chosen and how did the team verify that the microheater was actually operating at 350 C?
There should also be replicates demonstrated - can the authors generate more than one sample with this sensing behavior? It seems data from only one device (of each annealing type) is shown.
In Figure 6, why is the XRD of the U30 sensor shown twice? Is the U30 data in (a) the same sample as (b) prior to secondary heat treatment? Given likely variability in sample production and the small changes in crystallite size, I would like to see XRD of the same sample before and after secondary heat treatment. It is not clear if this is what is being shown. What k shape factor is used in calculating the Crystallite size in Table 1?
Overall, more details about needed about experimental conditions - gas pressures during annealing, gas flow rates during sensing, chemical concentrations during ultrasonic treatment, etc.
Reviewer 2 Report
The presented manuscript is far from being a research article, the reported work has serious academic and methodological gaps and is not ready for publication.
Here are some most important notes:
- Abstract describes experimental part, not the summary of work. Need to be rewritten.
- English usage requires significant improvement.
- Line 55 – what is charge aggregation?
- Line 129 – the conclusions on surface area based on SEM are questionable.
- What is the chemical process behind the nanorod growth?
- Lines 138-146 – no need to explain, well known phenomena, just giver reference
- Line 136 – what is recovery? How is it calculated? What does it characterize?
- Line 156 – are these values statistically significant& no error values are given to any calculated numbers.
- Line 158 – a conclusion is made with the aid of surface area, however the value is unknown
- Line 161-164 – the explanation is very primitive, supposedly the diffusion and adsorption phenomena are described, however with no experimental or literature basis. This is a prohibitive speculation.
- Line 171-186 – same situation, how heat treatment is connected with electron injection? Injection into what? What is recovery properties?
- Line 189-200 – same, how the XRD data can help to assume the grain size of anisotropic structures? How the grain size is connected with the “electron penetration into the depleted layer”?
- Table 1 – how the grain size of nanorods was calculated?
- The paper lacks sensor measurements to make conclusions on repeatability, stability. No data on other pollutants, humidity etc.
Reviewer 3 Report
Sensors based on ZnO nanonanorods were prepared to detect toluene in this work and the sensitivity was improved. However there are shortcomings of unclear logic and lack of innovation, as well as formatting and grammatical errors.
The specific issues are as follows:
- There are many formatting errors, such as punctuation, units and references, as well as many grammatical errors, please check carefully and correct them.
- I'm confused about the meaning of the paragraph on page 3 of 103. U5, U15, U20, U25 and U30 are five standard toxic gas types?
- (1) Clearer ruler is needed in Fig. 3. (2) Higher multiples of SEM and HRTEM are needed to clearly explain the increase of the length and size of ZnO nanorods. (3) The increase of specific surface area needs to be characterized by BET.
- Please explain the meaning of recovery (%) of ZnO, and define response time and recovery time in the article.
- Please add the response (as well as Ra) trend of all sensors to toluene with temperature and selectivity to other gases at the optimal operating temperature.
- Comparison with other toluene sensors should be added to highlight the advantages of this work.
- How to calculate the response of repeatability and concentration gradient while the sensor cannot recover to the original baseline. And as can be seen from Fig. 4, it is obvious that the baseline of the latter several times has reached about 10%, which is an impact that cannot be ignored for the responses (80%).
Round 2
Reviewer 1 Report
Concerns have been adequately addressed.
Author Response
We really appreciate your careful and favorable comments on our manuscript .
I'd like to thank you to give us a chance for improving this.
Reviewer 3 Report
The changes to the manuscript meet the reviewers’ requests. So, I recommend the manuscript be accepted for publication.
Author Response

(The authors gave the same response as above.)
